# Cationicity Enhancement on the Hydrophilic Face of Ctriporin Significantly Reduces Its Hemolytic Activity and Improves the Antimicrobial Activity against Antibiotic-Resistant ESKAPE Pathogens

**DOI:** 10.3390/toxins16030156

**Published:** 2024-03-18

**Authors:** Xudong Luo, Huan Deng, Li Ding, Xiangdong Ye, Fang Sun, Chenhu Qin, Zongyun Chen

**Affiliations:** 1Institute of Biomedicine and Hubei Key Laboratory of Embryonic Stem Cell Research, College of Basic Medicine, Hubei University of Medicine, Shiyan 442000, China; luoxudong000@126.com (X.L.); dh15736090032@163.com (H.D.); yexiangdong7237@163.com (X.Y.); 2016202040055@whu.edu.cn (F.S.); 2016202040025@whu.edu.cn (C.Q.); 2Department of Clinical Laboratory, Dongfeng Hospital, Hubei University of Medicine, Shiyan 442000, China; dl2168@163.com

**Keywords:** scorpion venom, short cationic antimicrobial peptide family, Ctriporin, peptide design, hemolytic activity, multidrug-resistant, ESKAPE

## Abstract

The ESKAPE pathogen-associated antimicrobial resistance is a global public health issue, and novel therapeutic strategies are urgently needed. The short cationic antimicrobial peptide (AMP) family represents an important subfamily of scorpion-derived AMPs, but high hemolysis and poor antimicrobial activity hinder their therapeutic application. Here, we recomposed the hydrophilic face of Ctriporin through lysine substitution. We observed non-linear correlations between the physiochemical properties of the peptides and their activities, and significant deviations regarding the changes of antimicrobial activities against different bacterial species, as well as hemolytic activity. Most importantly, we obtained two Ctriporin analogs, CM5 and CM6, these two have significantly reduced hemolytic activity and more potent antimicrobial activities against all tested antibiotic-resistant ESKAPE pathogens. Fluorescence experiments indicated they may perform the bactericidal function through a membrane-lytic action model. Our work sheds light on the potential of CM5 and CM6 in developing novel antimicrobials and gives clues for optimizing peptides from the short cationic AMP family.

## 1. Introduction

Antibiotic resistance is an ever-increasing threat to public health around the world. Of the most importance, the ESKAPE pathogens, namely *Enterococcus faecium*, *Staphylococcus aureus*, *Klebsiella pneumoniae*, *Acinetobacter baumannii*, *Pseudomonas aeruginosa*, and *Enterobacter species*, have received the most significant concern [1,2,3]. In this context, there is an urgent need to develop novel antimicrobials to combat these pathogens.

Cationic α-helical antimicrobial peptides (CαAMPs) have been largely identified from a wide range of organisms and reported to possess bactericidal activity against different kinds of pathogens [4,5,6,7,8,9]. CαAMPs are cationic amphiphilic molecules with a distinct hydrophilic face on which polar residues often position and a hydrophobic face where nonpolar residues often gather. Although the accurate mechanism for CαAMPs’ bactericidal activity has yet to be determined, a proposed mechanism was reported that CαAMPs could quickly kill the target bacterial cells through the formation of various kinds of lipophilic pores in the plasma membrane, resulting in metabolites leaking and finally cell death. Accordingly, CαAMPs exert their antimicrobial function with a significant reduction, although it may not be completely eliminated, of the probability of drug-resistant bacteria [10,11,12,13]. Therefore, CαAMPs represent one of the promising solutions for the treatment of antibiotic-resistant bacterial infections [14].

A wealth of CαAMPs have been identified from the venom glands of various scorpion species [4,9,15,16]. Among these peptides, the members of short cationic AMP family are a group of cationic α-helical peptides with 17 to 19 residues, including Ctriporin [17], Mucroporin [18], Imcroporin [19], Stigmurin [20], TsAP-1 and TsAP-2 [21], AamAP1 and AamAP2 [22], and BmKb1 and BmKb2 [23]. Some peptides from this family were reported to show antimicrobial activity against methicillin-resistant *S. aureus* [17,18,19,20]. We identified Lausporin-2 from the venom of *Liocheles australasiae*, displaying antimicrobial activity against several species of methicillin-resistant staphylococci, including *S. aureus*, *S. epidermidis* and *S. capitis* [24]. In a recent study, we revealed a moderate antimicrobial activity of Ctriporin against carbapenem-resistant (CRE) *A. baumannii* [25]. However, two main issues hinder the medicinal application of the peptides from short cationic AMP family: i, the peptides of this family were not reported to be active against other members of ESKAPE pathogens; ii, they have poor antimicrobial activity and potent hemolytic activity as reported in the previous studies.

In the present study, we chosen scorpion peptide Ctriporin as a scaffold template, tested its antimicrobial potential against ESKAPE pathogens; to improve its anti-ESKAPE potency, we recomposed the hydrophilic face of Ctriporin by gradually increasing the number of lysine residues on this surface. We found that compared with the parent peptide Ctriporin, the antimicrobial activities of two Ctriporin analogs, that are CM5 and CM6, against antibiotic-resistant ESKAPE pathogens were significantly improved, while their hemolytic activities were significantly reduced. We observed non-linear correlations between the physiochemical properties of the peptides and their activities, and significant deviations regarding the changes of antimicrobial activities against different bacterial species, as well as their hemolytic activity. Mechanism studies suggested that the peptides may kill the bacteria through a membrane-lytic mode of action. Our study provided new molecules for drug development against ESKAPE pathogen infections; this study also provides insight into the optimization of other scorpion-derived CαAMPs, especially those of short cationic AMP family.

## 2. Results

### 2.1. Analogue Design of Ctriporin

Ctriporin is a 19-mer CαAMP identified from the scorpion *Chaerilus tricostatus* [17]. The HeliQuest server is an online platform that facilitates the analysis of the helix properties of a certain CαAMP [26]. First, we generated the helical wheel plot of the peptide using the HeliQuest server. As shown in Figure 1A, Ctriporin presents obvious amphiphilic characteristics in that most of the hydrophilic residues are located on one side (the hydrophilic face) and most of the hydrophobic residues on the other side (the hydrophobic face). To improve the antimicrobial potency of Ctriporin, we focused on the hydrophilic face of the peptide, and designed a series of analogs, namely CM1 to CM7, by gradually replacing the non-hydrophobic residues with lysine (K). Note that, since proline was shown to play a crucial role in determining CαAMP’s performance [27], the proline at the seventh site of the peptide (Pro^7^) is the last residue to be replaced in our study. The sequences and the physiochemical parameters of Ctriporin and its analogs are shown in Table 1. With the increase of the number of lysine, the net charge of the peptide increases from +3 to +10, and the hydrophobic moment of the peptides increases gradually from 0.452 to 0.704, while the hydrophobicity of these peptides decreases gradually from 0.805 to 0.377. As can be seen in Appendix A, the increase of net charge on the hydrophilic face of Ctriporin induced a nearly linear changes of hydrophobicity and hydrophobic moment of the peptide, indicating that the increase of net charge is a useful approach to change the hydrophobicity and hydrophobic moment of the peptide.

CαAMPs display α-helical conformation in plasma membrane mimic environments. Therefore, the secondary structural contents of the peptides were measured using a circular dichroism (CD) Spectrometer. As shown in Figure 1B, the CD spectra of Ctriporin and its analogs exhibited a negative peak at a wavelength near 200 nm, indicating random coiled conformations of these peptides. However, as shown in Figure 1C, the CD spectra of Ctriporin and its analogs exhibited a positive peak at a wavelength near 195 nm and two negative peaks at 208 and 222 nm, indicating typical α-helical conformations. Note, compared with other peptides, CM7 exhibited a significant different CD spectra as the intensity of the negative peak at 222 nm was significantly reduced, indicating that the substitution of Pro^7^ may have impact on the secondary structure of this peptide in membrane mimic environment.

### 2.2. CM5 and CM6 Exhibited the Most Potent Antimicrobial Activity and the Lowest Hemolytic Activity

The ESKAPE pathogens represent the main threat to public health concerning antimicrobial resistance. Therefore, we first evaluated the antimicrobial activities of Ctriporin and its analogs against the standard ESKAPE strains. The minimum inhibitory concentration (MIC) was chosen as the index for the evaluation of each peptide’s activity [28]. As shown in Table 2, the MICs of Ctriporin against *S. aureus* and *A. baumannii* were determined to be 16–32 μg/mL, and for other tested bacterial species, these values were ≥64 μg/mL. Compared with Ctriporin, increasing the number of lysine on the hydrophilic face has a significantly positive effect on its antimicrobial activity. As can be seen, the MICs of CM3 and CM6 against these pathogens were determined to be 4–8 μg/mL and those for CM5 were determined to be 2–8 μg/mL. Note that the MICs of CM4 and CM7 are 4–32 and 8–32 μg/mL, respectively, indicating the existence of significant bacterial species deviations for the antimicrobial activity of these peptides. In addition, the replacement of specific residues, especially Pro^7^, may harm the antimicrobial activities of the peptide. Taken together, CM3 (4–8 μg/mL), CM5 (2–8 μg/mL), and CM6 (4–8 μg/mL) are peptides showing the most potent activities against the ESKAPE pathogens and the smallest bacterial species deviation.

Another purpose of this work is to attenuate Ctriporin’s hemolytic activity. As shown in Figure 2, the curves of hemolysis% vs. peptide concentration of CM1, CM2, and CM3 are similar to that of Ctriporin. Interestingly, the percent hemolysis of CM4, CM5, and CM6 decreased dramatically compared with Ctriporin. Like antimicrobial activity, the replacement of Pro^7^ harms hemolytic activity attenuation, as the curve of hemolysis% of CM7 positions closely to those of CM3. To better illustrate the difference in each peptide’s hemolytic activity, the hemolysis% values of these peptides at 256 μg/mL were summarized in Table 2. As can be seen, CM5 (9.8 ± 0.6%) and CM6 (8.1 ± 1.9%) are the two peptides with the lowest hemolytic activity. Although CM3 displays potent MICs (4–8 μg/mL), it also shows strong hemolytic activity. Therefore, CM5 and CM6 are the two Ctriporin analogs that exhibit the most potent antimicrobial activity and the lowest hemolytic activity. 

### 2.3. The Correlations between the Physiochemical Properties and the Activities of Ctriporin and Its Analogs

The physicochemical properties, that are the net charge, the hydrophobicity, and the hydrophobic moment, are three critical parameters that affect CαAMPs’ activities. However, the correlations between the physiochemical properties of CαAMPs and their antimicrobial or hemolytic activities are complicated and vary significantly between different CαAMPs [29]. In an attempt to provide clear profiles between these parameters and the activities of Ctriporin and its analogs investigated in this study, Figure 3 was generated through locally weighted regression fitting [30,31]. Generally, non-linear correlations were observed between physicochemical properties and the functions of these peptides. In terms of antimicrobial activity, bowl-shaped curves were obtained. In detail, as the net charges and the hydrophobic moments increase, and the hydrophobicity decreases from Ctriporin to CM5, either the values of MICs against *S. aureus* (Figure 3A–C) or the values against *A. baumannii* (Figure 3D–F) decrease rapidly, indicating the improvement of antimicrobial activities; however, further changes in the same directions led to increase of MICs, indicating the damage to antimicrobial activities. In addition, Figure 3A–F displays distinct sharps of *S. aureus* vs. *A. baumannii*, combining the results shown in Table 2, these results indicated significant variations in MICs between bacterial species.

In terms of hemolytic activity, however, as shown in Figure 3G–I, curves with “Z” or “S” sharps were observed. No significant changes in hemoysis% were observed when the net charges increased from +3 to +6, the values of hydrophobicity decreased from 0.805 to 0.651, or the hydrophobic moments increased from 0.452 to 0.592. After that, the percentage of hemolysis decreased quickly; however, a significant increase in the percentage of hemolysis was observed in CM7. Taken together, our analysis suggested non-linear correlations between the physiochemical properties and the activities of Ctriporin and its analogs designed in this study, and significant variations between bacterial species, as well as hemolytic activity vs. antimicrobial activities.

### 2.4. Antimicrobial Activities of CM5 and CM6 against Clinical Isolates of ESKAPE Pathogens

To further evaluate the antimicrobial potential of CM5 and CM6, the MICs of these two peptides, as well as the parent peptide Ctriporin, were determined against clinical antibiotic-resistant isolates of ESKAPE strains. As shown in Table 3, Ctriporin displayed poor antimicrobial activities against these strains, whereas those of CM5 and CM6 were significantly improved. In particular, CM5 and CM6 displayed MIC values of 2 μg/mL against clinical isolates of CRE and multidrug-resistant (MDR) *A. baumannii*. Note that CRE and MDR *A. baumannii* are listed at the top of the pathogens threatening public health [32]. Compared with the parent peptide Ctriporin, eight to sixteen times improvement in terms of antimicrobial activity was obtained through our peptide design strategy.

### 2.5. Growth Inhibitory Effects of CM5

To further confirm the antimicrobial activity of the peptides, CM5 was chosen and the growth inhibitory curves were measured against *A. baumannii* ATCC19606, and the clinical isolates 19295, 19618, and 19462. As shown in Figure 4, compared with the samples without treatment of CM5, the additions of 0.25 and 0.5 × MIC of the peptide have no inhibitory effects on the growth of the strains ATCC19606, and the clinical isolates 19295 and 19618. Still, the same dosages displayed some extent of inhibitory effects on the clinical isolates 19462. Interestingly, a similar phenomenon has been observed for the same strain in our previous study [25], this feature may imply some unique feature of this strain, or provide some clues as to the mechanism of action of the peptides, which deserves further investigation in future studies. Finally, the additions of 1, 2, and 4 × MIC of the peptide inhibited the growth of all tested strains, which is consistent with the results of MIC assays.

### 2.6. Bacterial-Killing Kinetics

To evaluate the bacterial-killing ability of CM5, *A. baumannii* ATCC19606 was chosen and the time-killing kinetics of the peptide against this bacterial strain was measured. As shown in Figure 5, a concentration-dependent decrease of viable bacterial cells was observed in a short time interval upon the addition of 1, 2, and 4 × MIC of the peptide. The results indicated that CM5 exhibits fast bactericidal activity. 

### 2.7. CM5 Induces Dose-Dependent Membrane Disruptions of the Bacterial Cells

It has been suggested that linear CαAMPs kill pathogens through pore formation in the plasma membrane of targeted bacterial cells, that is the so-called membrane-lytic mode of action [8,10]. PI is a classical DNA-binding probe that can permeate only compromised cell membranes [33]. It is a useful tool for the detection of membrane disruptions caused by CαAMPs. As shown in Figure 6A,B, the co-incubation of CM5 with *A. baumannii* ATCC19606 caused fast and dose-dependent PI fluorescence increase, indicating the addition of the peptide induced membrane disruption on the plasma membrane of the bacterial cells.

For further demonstration of this effect, the SYTO™ Green fluorescence assay was performed. SYTO™ Green is another DNA-binding probe displaying the same permeation mechanism as PI but with a different chemical structure and property, and was often used to illustrate membrane disruptions induced by antimicrobial peptides [34]. As shown in Figure 6C,D, a similar dose-dependent fluorescence increase was observed for the bacterial cells incubated with CM5. Taken together, the fluorescence assays, combined with time-killing kinetics assay, suggested that CM5 may perform a membrane-lytic mode of action.

## 3. Discussion

The ESKAPE pathogens exert a significant threat to public health. To fight against ESKAPE-caused antimicrobial resistance, the development of novel antimicrobials is of the most significance. Ctriporin is a highly hemolytic linear cationic antimicrobial peptide that has been demonstrated to be moderately active against bacteria *S. aureus* and *A. baumannii*, but poorly or inactive against other kinds of ESKAPE pathogens [15,25]. In this study, through cationicity enhancement on the hydrophilic face of the peptide by lysine substitution, we obtained CM5 and CM6, two Ctriporin analogs with significantly reduced hemolytic activity and much improved antimicrobial activities against all tested ESKAPE strains. Our work provided new clues for the development of antimicrobials against infectious diseases caused by ESKAPE pathogens. 

The high hemolytic activity and low antimicrobial activity represents the major disadvantages of natural CαAMPs. Natural CαAMPs are characterized by the relatively high occurrence frequency of alkaline residues such as lysine and arginine [35]. Although there are exceptions, in most cases, the alkaline residues are positioned on the hydrophilic surface of the peptides. These two kind of residues were used in several previous studies regarding natural CaAMPs in an attempt to reduce the hemolytic activity and improve their antimicrobial activities against different bacterial species [36,37,38,39,40,41,42,43]. However, due to the extreme diversity of sequence composition of these peptides, the above-mentioned strategies were largely based on empirical trial and error, and may not fit for other CαAMPs’ optimization. In our previous studies, we demonstrated the improvement of the antimicrobial potency of 13-mer scorpion-derived CαAMP BmKn2 through cationicity enhancement, followed by replacing arginine with lysine on the hydrophilic face of the peptide, and designed BmK2-7K, a CαAMP composed of sole lysine on its hydrophilic face, displaying relative low hemolytic activity and potent antimicrobial activities against antibiotic-resistant ESKAPE pathogens [44,45]. To further investigate lysine replacements on the hemolytic and antimicrobial activities of natural peptides from another family of scorpion-derived CαAMP, we focused on the hydrophilic surface of scorpion-derived 19-mer peptide Ctriporin belonging to short cationic antimicrobial peptide family and found that increasing the number of lysine on this surface led to significant hemolysis attenuation and antimicrobial activity enhancement. Our studies indicated that lysine substitutions on the hydrophilic face may be a useful strategy for scorpion-derived short CαAMPs. However, this study revealed distinct optimal profiles for net charge adjustment of the peptides regarding different bacterial species. In addition, the optimal net charge range of the peptides regarding hemolytic activity was observed to be more narrow compared with that of antimicrobial activities in most cases. These results indicated the difficulty when designing low-toxic broad-spectrum antimicrobial peptides. Finally, unlike 13-mer CαAMP BmKn2, excessive increases of lysine were harmful to Ctriporin analogs. This phenomenon was also observed in the previous study [46], however, in our case, is this result caused by proline mutation which led to significant structural change in the peptide, as evidenced by our CD spectra profile needs further investigation.

To date, several groups of scorpion-derived CαAMPs have been identified so far from many different scorpion species [4]. However, the previous investigations were mostly focused on the identification and optimization of short peptides (i.e., 13–14 amino acids) [42,43,45,47,48,49,50], and the therapeutic potential of peptides of short cationic antimicrobial peptide family was largely overlooked due to their high hemolytic activity and poor antimicrobial activity. We for the first time performed a structure-activity relationship investigation on Ctriporin and obtained two analogs, CM5 and CM6. They were demonstrated to display much lowered hemolytic activity and much better antimicrobial activities against antibiotic-resistant ESKAPE pathogens. Our study sheds light on the therapeutical potential of the members of the short cationic antimicrobial peptide family and provides clues for the optimization of these peptides.

## 4. Materials and Methods

### 4.1. Peptide Analysis and Synthesis

All peptides used in this study were synthesized using solid-phase methods by Sangon Biotech (Shanghai, China) and amidated at the C-terminus of the peptides. The purities of the peptides (>95%) were analyzed by reversed-phase high-performance liquid chromatography [51]. The molecular weights of the peptides were measured by mass spectrometry. The helical-wheel plots, net charges, mean hydrophobicity values, and hydrophobic moments of the peptides were obtained using the HeliQuest server (https://heliquest.ipmc.cnrs.fr/, accessed on 5 March 2024) [24,26].

### 4.2. Circular Dichroism

The secondary structural contents of the peptides were measured at 25 °C using a circular dichroism spectrometer (Chirascan plus, Applied Photophysics, Charlotte, NC, USA). A quartz cell with a 0.1 cm light path was employed in the experiments. Peptides were dissolved in either distilled H_2_O or 30 mM sodium dodecyl sulfate (Sigma-Aldrich, St. Louis, MO, USA) solutions to achieve a concentration of 150 μg mL^−1^. Three scans of the CD spectra (λ_190–250_ nm) were obtained for each sample. The mean residue molar ellipticity was calculated from the original CD data by the equation, as described in the previous study [44]. The results are the average ± standard error of data derived from three independent experiments.

### 4.3. Bacterial Strains

The standard strains, including *S. aureus* ATCC29213 and ATCC25923, *E. faecalis* ATCC29212, *E. coli* ATCC25922 and ATCC35218, *A. baumannii* ATCC19606, *K. pneumoniae* ATCC700603, and *P. aeruginosa* ATCC27853, were purchased from the China Center of Type Culture Collection (CCTCC). The clinical strains of the ESKAPE pathogens were isolated from the affiliated hospitals of Hubei University of Medicine [24,25,44]. All of them were stored at −80 °C before use. The drug sensitivities of the clinical strains were determined by the Kirby–Bauer test [52].

### 4.4. Minimum Inhibitory Concentration (MIC) Determination

The MICs were determined using the broth dilution method according to the guidelines of the Clinical and Laboratory Standards Institute (CLSI), as described in the previous studies [44,45]. Briefly, the bacterial cells were incubated in Mueller–Hinton broth (MHB, Oxoid, Basingstoke, UK) at 37 °C and 150 rpm to the mid-growth phase and diluted to achieve a final concentration of 5 × 10^5^ colony-forming units per milliliter (CFU/mL). Then, the cells were mixed with each peptide in different concentrations (1, 2, 4, 8, 16, 32, and 64 μg/mL). After co-incubation in MHB (150 rpm; 37 °C) for 16–20 h, the optical density at 630 nm (OD_630_) was determined, and the MIC value of each peptide was defined as the minimum peptide concentration with no detectable absorbance increase. All experiments were repeated at least three times.

### 4.5. Hemolytic Activity Determination

The hemolytic activities of the peptides were determined against human red blood cells (hRBCs). As described [44], sodium citrate-anticoagulated hRBCs were washed three times with 0.9% sodium chloride and diluted to a final concentration of 4% (*v*/*v*). Then, an equal volume of hRBC solutions was mixed with an equal volume of each peptide in a series of concentrations (0, 2, 4, 8, 16, 32, 64, 128, 256, and 512 μg/mL). After co-incubation at 37 °C for 1 h, the degree of hemolysis of the peptides was determined by detecting the release of hemoglobin from the damaged hRBCs by measuring the optical density at 540 nm (OD_540_). hRBCs in 0.9% sodium chloride and 1% Triton X-100 were considered 0% and 100% hemolysis, respectively. The results shown are the average ± standard error of data derived from three independent experiments. The percent hemolysis was calculated according to the following equation [53,54]:Hemolysis (%) = 100 × (A_sample_ − A_blank_)/(A_positive_ − A_blank_)

### 4.6. Growth Curve

The growth inhibitory effects of the peptides were measured against the standard strain *A. baumannii* ATCC19606 and the corresponding clinical isolates 19295, 19618, and 19462. Bacterial cells (5 × 10^5^ CFU/mL) were incubated in MHB (150 rpm; 37 °C) without or with 0.25, 0.5, 1, 2, and 4 × MIC of the peptide for 9 h. At different time intervals, the OD_630_ was measured to produce the curves of bacterial growth vs. time. The results are the average ± standard error of data derived from three independent experiments.

### 4.7. Time-Killing Kinetics

Time-killing kinetics of the peptide were determined through the plate counting method. Briefly, 2 × 10^6^ CFU/mL bacterial cells were incubated in MHB at 150 rpm and 37 °C without or with the peptide at different concentrations (1, 2, and 4 × MIC). At each time interval (0, 5, 15, 30, and 60 min), aliquots were taken, and diluted serially, and 100 μL of the sample was spread on Mueller–Hinton agar. After overnight incubation, the surviving bacterial colonies of each plate were counted. The results are the average ± standard error of data derived from three independent experiments.

### 4.8. Membrane Permeabilization Assay

Plasma membrane permeabilization of the bacterial cells caused by the peptide was determined by propidium iodide (PI, Thermo Fisher, Waltham, MA, USA) and SYTO™ Green (Thermo Fisher, Waltham, MA, USA) uptake assays, as described previously [25]. Briefly, exponential phase bacterial cells were collected by centrifugation (6000× *g*, 4 °C) and washed with phosphate-buffered saline (PBS). Next, the bacterial cells were diluted with PBS to achieve an absorbance of 0.1 at 630 nm and prepared in a sterile 96-well polypropylene plate. After mixing with 2 μM PI or 30 nM SYTO™ Green, a series dose of the peptide (1, 2, 4, and 8 × MIC) was added into each well, and fluorescence was immediately recorded on a Molecular Devices SpectraMax i3x (excitation wavelength: 535 nm for PI and 503 nm for SYTO™ Green; emission wavelength: 617 nm for PI and 530 nm for SYTO™ Green). All experiments were repeated at least three times. The results are the average ± standard error of data derived from three independent experiments.

## Figures and Tables

**Figure 1 toxins-16-00156-f001:**
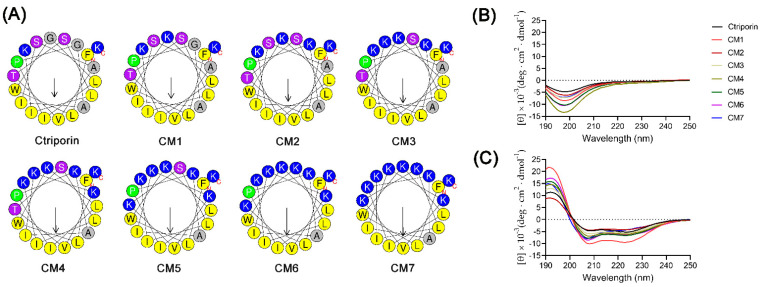
Structural analysis of Ctriporin and its analogs. (**A**) The helical wheel diagrams. The hydrophobic residues and lysine are presented in yellow and blue color, respectively. (**B**,**C**) CD spectra of Ctriporin and its analogs (150 μg mL^−1^) in ddH_2_O and 30 mM sodium dodecyl sulfate (SDS) solutions.

**Figure 2 toxins-16-00156-f002:**
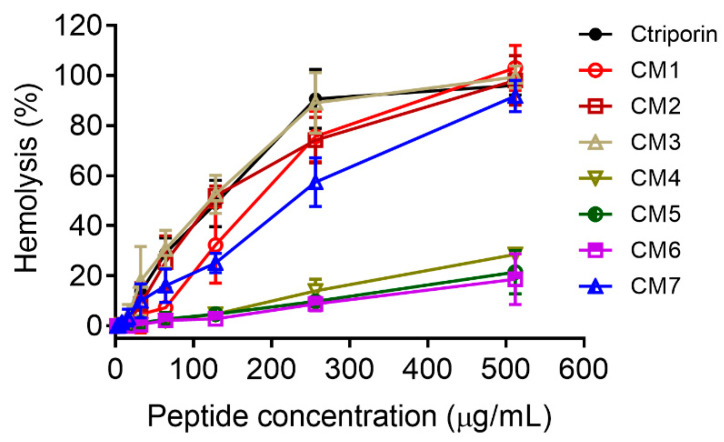
Hemolytic activities of Ctriporin and its analogs against human red blood cells (hRBC). Hemolytic activities of Ctriporin (solid circle), CM1 (open circle), CM2 (open square), CM3 (upward triangle), CM4 (downward triangle), CM5 (half solid circle), CM6 (half solid square), and CM7 (half solid triangle), were measured against 2% (*v*/*v*) human red blood cells. The peptide concentrations are 0, 2, 4, 8, 16, 32, 64, 128, 256, and 512 μg/mL. The cells treated with 1% Triton X-100 were treated as 100% hemolysis. All experiments were repeated three times, and the data shown are the average ± standard error.

**Figure 3 toxins-16-00156-f003:**
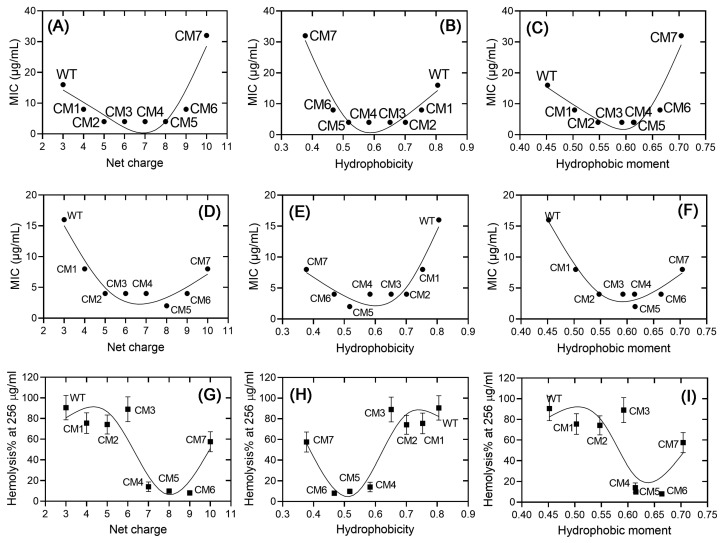
The correlations between the net charge (**A**,**D**,**G**), the hydrophobicity (**B**,**E**,**H**), the hydrophobic moment (**C**,**F**,**I**) of the peptides and their activities. (**A**–**C**): *S. aureus* ATCC29213; (**D**–**F**): *A. baumannii* ATCC19606; (**G**–**I**): hemolysis% of the peptides at 256 μg/mL. Note, the hydrophobic moments of CM4 and CM5 are very close, the nodes are overlapped in (**C**). The error bar represents the standard variation of each value.

**Figure 4 toxins-16-00156-f004:**
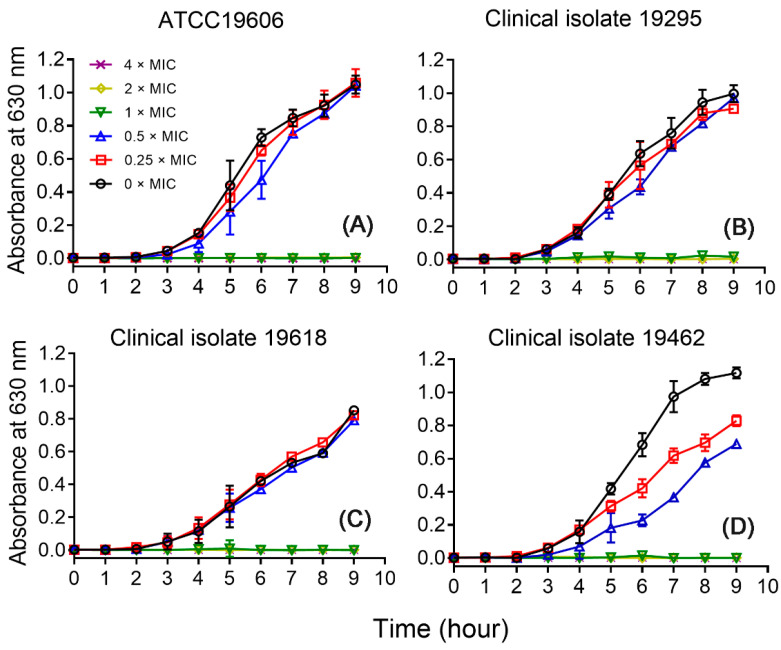
Time-growth curves of CM5 treated *A. baumannii* ATCC19606 (**A**), and the clinical isolates 19295 (**B**), 19618 (**C**), and 19462 (**D**). The peptide concentrations were 0 (open circle), 0.25 (open square), 0.5 (upward triangle), 1 (downward triangle), 2 (diamond), and 4 (cross) × MIC. The curves shown are the average ± standard error of data derived from three independent experiments.

**Figure 5 toxins-16-00156-f005:**
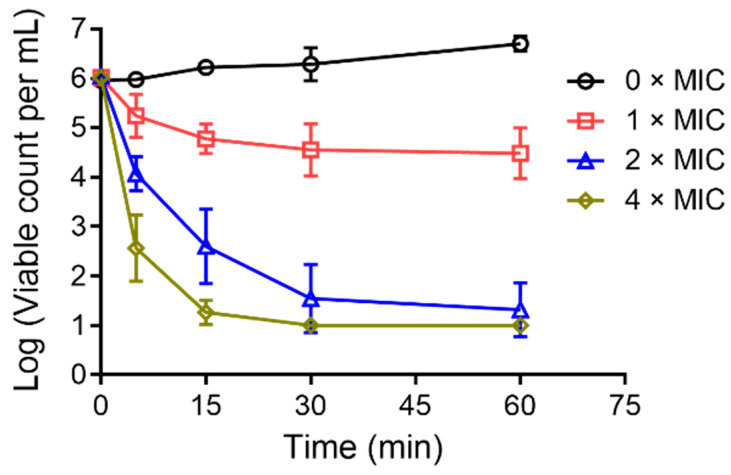
Time-killing kinetics of *A. baumannii* ATCC19606 treated with CM5. The bacterial cells were incubated without (open circle) or with 1 (open square), 2 (upward triangle), and 4 (diamond) × MIC peptides, respectively. The data shown are the means ± SD of three independent experiments.

**Figure 6 toxins-16-00156-f006:**
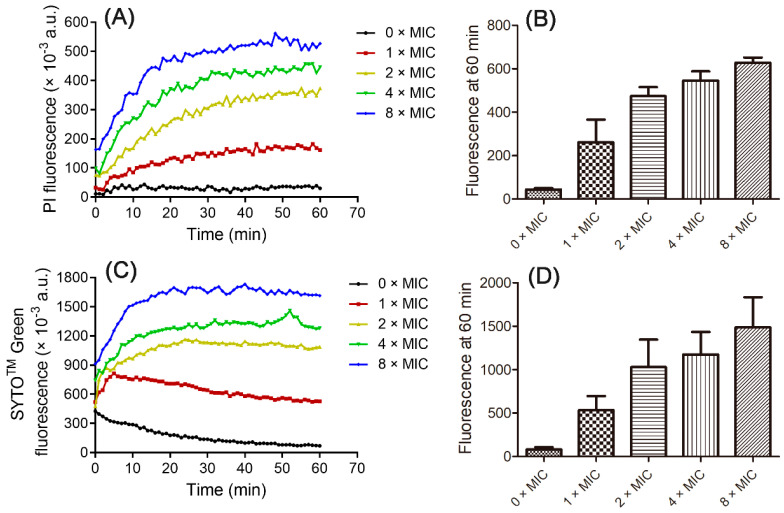
Membrane permeation of *A. baumannii* ATCC19606 treated with CM5. (**A**) PI fluorescence kinetics of the bacterial cells treated with the peptide; (**B**) PI fluorescence at 60 min; (**C**) SYTO™ Green fluorescence kinetics of the bacterial cells treated with the peptide. (**D**) SYTO™ Green fluorescence at 60 min. In (**A**,**C**), the bacterial cells incubated without (black) or with 1 (brown), 2 (yellow), 4 (green), and 8 (blue × MIC are shown, respectively. The data shown are the means ± SD of three independent experiments.

**Table 1 toxins-16-00156-t001:** Physiochemical parameters of Ctriporin and its analogs.

Peptide	Sequence *^a^*	aa *^b^*	MW *^c^*	z *^d^*	<H> *^e^*	<μH> *^f^*
Ctriporin	FLW**G**LI**PG**AI**SA**V**TS**LIKK	19	2013.5	3	0.805	0.452
CM1	FLW**K**LIPGAISAVTSLIKK	19	2084.6	4	0.753	0.503
CM2	FLW**K**LIP**K**AISAVTSLIKK	19	2155.7	5	0.701	0.547
CM3	FLW**K**LIP**K**AI**K**AVTSLIKK	19	2196.8	6	0.651	0.592
CM4	FLW**K**LIP**K**AI**KK**VTSLIKK	19	2253.9	7	0.583	0.614
CM5	FLW**K**LIP**K**AI**KK**V**K**SLIKK	19	2282.0	8	0.517	0.615
CM6	FLW**K**LIP**K**AI**KK**V**KK**LIKK	19	2323.1	9	0.467	0.664
CM7	FLW**K**LI**KK**AI**KK**V**KK**LIKK	19	2353.1	10	0.377	0.704

*^a^* All peptides are amidated at the C-terminus; *^b^* aa, amino acids; *^c^* MW, the molecular weights were determined with mass spectrometry; *^d^* z, the net charges were determined at pH 7.4; *^e^* <H>, the mean hydrophobicity; *^f^* <µH>, the hydrophobic moments.

**Table 2 toxins-16-00156-t002:** Antimicrobial activities of Ctriporin and its analogs against standard strains.

	*S. aureus* ATCC29213	*S. aureus* ATCC25923	*E. faecium* ATCC29212	*E. coli* ATCC25922	*E. coli* ATCC35218	*P. aeruginosa* ATCC27853	*K. pneumoniae* ATCC700603	*A. baumannii* ATCC19606	
	MICs μg/mL (μM)								Hemolysis% at 256 μg/mL
Ctriporin	16 (8.0)	32 (15.9)	64 (31.8)	64 (31.8)	>64 (31.8)	>64 (31.8)	>64 (31.8)	16 (8.0)	90.6 ± 11.8
CM1	8 (3.8)	16 (7.7)	16 (7.7)	8 (3.9)	32 (15.4)	>64 (30.7)	64 (30.7)	8 (3.8)	75.6 ± 10.1
CM2	4 (1.9)	8 (3.7)	4 (1.9)	4 (1.9)	8 (3.7)	16 (7.4)	16 (7.4)	4 (1.9)	74.3 ± 9.2
CM3	4 (1.8)	4 (1.8)	4 (1.8)	4 (1.8)	4 (1.8)	8 (3.6)	8 (3.6)	4 (1.8)	89.1 ± 12.1
CM4	4 (1.8)	8 (3.6)	16 (7.1)	8 (3.6)	16 (7.1)	16 (7.1)	32 (14.2)	4 (1.8)	14.0 ± 4.7
CM5	4 (1.8)	4 (1.8)	8 (3.5)	8 (3.5)	4 (1.8)	4 (1.8)	8 (3.5)	2 (0.9)	9.8 ± 0.6
CM6	8 (3.4)	8 (3.4)	8 (3.4)	8 (3.4)	8 (3.4)	4 (1.7)	8 (3.4)	4 (1.7)	8.1 ± 1.9
CM7	32 (13.6)	32 (13.6)	32 (13.6)	32 (13.6)	16 (6.8)	8 (3.4)	32 (6.8)	8 (3.4)	57.5 ± 9.8

**Table 3 toxins-16-00156-t003:** Antimicrobial activities of Ctriporin, CM5 and CM6 against the clinical strains of the ESKAPE pathogens.

Strains	Resistance	Ctriporin	CM5	CM6
		MICs μg/mL (μM)		
*S. aureus* 9124	MRSA *^a^*	8 (4.0)	8 (3.5)	8 (3.4)
*S. aureus* 891	MRSA	16 (8.0)	4 (1.8)	8 (3.4)
*S. epidermidis* 6943	MRSE *^b^*	8 (4.0)	4 (1.8)	4 (1.7)
*S. epidermidis* 9092	MRSE	8 (4.0)	4 (1.8)	4 (1.7)
*S. capitis* 3255	MRSC *^c^*	8 (4.0)	2 (0.9)	2 (0.86)
*E. faecium* 898	MDR *^d^*	16 (8.0)	4 (1.8)	4 (1.7)
*E. coli* 2678	ESBL *^e^*	64 (31.8)	8 (3.5)	8 (3.4)
*E. coli* 2687	ESBL	>64 (31.8)	8 (3.5)	8 (3.4)
*P. aeruginosa* 9014	CRE *^f^*/MDR	>64 (31.8)	8 (3.5)	8 (3.4)
*K. pneumoniae* 9126	CRE	>64 (31.8)	16 (7.0)	16 (6.9)
*A. baumannii* 19295	CRE/MDR	32 (15.9)	2 (0.9)	2 (0.86)
*A. baumannii* 19462	CRE/MDR	16 (8.0)	2 (0.9)	2 (0.86)
*A. baumannii* 19618	CRE/MDR	32 (15.9)	2 (0.9)	2 (0.86)

*^a^* MRSA: methicillin-resistant *S. aureus*; *^b^* MRSE: methicillin-resistant *S. epidermidis*; *^c^* MRSC: methicillin-resistant *S. capitis*; *^d^* MDR: multidrug resistant; *^e^* ESBL: Extended-spectrum β-lactamases; *^f^* CRE: carbapenem-resistant.

## Data Availability

The data used and/or analyzed during the current study is available from the corresponding author on reasonable request.

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
