# Peer review of "Cationicity Enhancement on the Hydrophilic Face of Ctriporin Significantly Reduces Its Hemolytic Activity and Improves the Antimicrobial Activity against Antibiotic-Resistant ESKAPE Pathogens"

_toxins, 2024, doi:10.3390/toxins16030156_

Round 1

Reviewer 1 Report

Comments and Suggestions for Authors

The manuscript entitled « Cationic enhancement on the hydrophilic face of Ctriporin significantly reduces its hemolytic activity and improves the antimicrobial activity against antibiotic-resistant ESKAPE pathogens » (toxins_2737304), looks at the antimicrobial activity of peptides derived from scorpion (Chaerilus tricostatus) venom. Ctiporin belongs to the aAMP family of cationic amphipathic molecules found in a wide range of organisms. In a previous study, the authors showed that the number of Lysine residues on the hydrophilic side of BmKn2-7, another aAMP, influence not only its antimicrobial properties but also its hemolytic activity. The authors extend these observations in the present manuscript: the increase in the number of Lys residues is accompanied by an increase in antimicrobial efficacy (estimated by the minimum inhibitory concentration, MIC) and a decrease in hemolytic activity (Tables 2 and 3). These data are confirmed by an evaluation of the bacterial growth of several strains (Fig. 5), and of the toxicity of one of the peptides on the survival of A. baumannii (Fig. 6). Finally, by using fluorescence assays, the authors suggest that Ctriporin-derived peptides might use a membrane-lytic mode of action.

Overall, this is an interesting study that focuses on a real public health problem, namely the development of multi-drug resistant bacteria. However, there are a number of details that, in my opinion, preclude publication of this manuscript in its current form. The main issue is the use of MICs, and the correlation that the authors have attempted to establish between MICs and the physico-chemical parameters for the various peptides.

MCI values cannot be averaged, even if they are defined values, and what about if they are undefined values (>64 microM means that the MCI value can be anything greater than 64 microM!).

Moreover, I do not understand what is remarkable about the fact that hydrophobicity decreases linearly as a function of the number of Lys residues, and conversely, that hydrophobic moment increases? The authors need to be more explicit on this specific point.

And since the number of Lys residues is linearly correlated with hydrophobicity, and hydrophobic moment, why is it surprising that one get the same graph three times when one plot these parameters against "averaged MCI" (even though this means nothing) or hemolysis% (Fig 7)? Although I understand the approach, averaged MCI cannot be used to establish a correlation between physico-chemical parameters and anti-microbial potency of the different peptides. This section must be modified or removed.

Minor comments

Line 85 “To improve the functional performance…” I guess it means anti-microbial potency?

Line 110 S. aureus… please use italic characters throughout

Line 154-155 If the sentence is an assertion, then references are needed.

Line 231 please correct “plasma”.

Line 223 I don’t understand why the authors use two similar (although not strictly identical) fluorescent assays? They both give same results. The reference [30] is not a study describing what differentiates PI and SYTO Green!

Reviewer 2 Report

Comments and Suggestions for Authors

This article explores the structure-activity relationship of Ctriporin, a scorpion-derived toxin comprising 19 amino acids. The substitution of certain amino acids with lysine has enhanced antimicrobial activity and reduced hemolytic activity. Understanding the physical and biological properties of the Ctriporin analogs, along with additional ones, is crucial for better understanding the nature of the lysine substitution strategy, the mechanisms of action and side effects of Ctriporin. Before publication in Toxins, the article requires revisions and additional experiments, as outlined below.

1.       The authors assume that Ctriporin analogs will share the same α-helix structure as the original Ctriporin. Experimental verification, such as measuring CD spectra for Ctriporin and its analogs, is needed.

2.       Is it necessary to make substitutions to lysine in all five amino acids to obtain biological activity like that of Ctri-M5? From the standpoint that certain amino acid substitutions, rather than the overall peptide properties, are important for obtaining ideal biological activities, one would expect G8K, A12K, and T19K to be important substitutions, and conversely, G4K, S11K, S15K, and P7K would be substitutions that need not necessarily be made. Determining the minimal lysine substitution required for ideal weak hemolytic and strong antibacterial activity, like in Ctri-M5, is crucial. Such experimental results for additional analogs will improve the lysine substitution strategy.

3.       Hemolytic activity of Ctri-M5 and Ctri-M6 at lower concentrations (e.g., 16 and/or 32 μg/mL) is essential for clinical relevance. Given their stronger antimicrobial activity, data at lower concentrations are more pertinent.

4.       The results presented in Figure 5 show that for the three bacteria in panels (A)-(C), Ctrin-M5 shows no antimicrobial activity at concentration of 0.5×MIC and only at concentrations above 1×MIC. Antimicrobial agents of low molecular weight chemical compounds show concentration dependence as shown in panel (D). An explanation of this characteristic concentration dependence of Ctri-M5 should be added. This feature may provide some clues as to the mechanism of action of Ctrin-M5 and Ctriporin.

5.       The Discussion section redundantly reiterates points from the Results and Introduction. It should be rewritten, emphasizing specific discussion points. Clarification on lysine substitution versatility and rules for determining substituted amino acids is essential.

6.       In Figures 2 and 4, clarity is needed to distinguish marks representing various Ctriporin analogs. Figure 2's legend should specify whether black open circles and blue open squares denote hydrophobic or hydrophilic moments, respectively.

7.       The Experimental Methods section should provide details on culture conditions for all bacteria used in the study.

8.       Statistical analysis methods for interpreting results from multiple-repeated experiments should be elucidated in the Experimental Methods section.

Comments on the Quality of English Language

None.

Reviewer 3 Report

Comments and Suggestions for Authors

The reviewer would like to thank the authors for their submission titled, “Cationicity Enhancement on the Hydrophilic Face of Ctriporin Significantly Reduces Its Hemolytic Activity and Improves the Antimicrobial Activity Against Antibiotic-Resistant ESKAPE Pathogens”.  In this article the authors hypothesize various aaàK substituted Ctriporins will exhibit improved antimicrobial activity and decreased hemolysis.  They attempt to show this through MIC assays, time kill assays, as well as fluorescent staining for membrane disruption.  The reviewer will offer a critique based on the Style aspects and the Technical aspects of the paper.

Style:

1.      While certainly membrane active peptides are important as a possible new class of antimicrobials, they, like many other drug classes, have drawbacks and limitations.  It might be worth mentioning at least in brief.  A potentially useful review would be:

Kmeck, A., Tancer, R. J., Ventura, C. R., & Wiedman, G. R. (2020). Synergies with and resistance to membrane-active peptides. Antibiotics, 9(9), 620. https://doi.org/10.3390/antibiotics9090620

2.      It may be helpful to provide a comparison of peptides in a micromolar ratio as well as in micrograms/mL.

Technical:

1.      Can the authors please provide a calculation for %helicity.  A good reference for this would

be:

 Luo, P., & Baldwin, R. L. (1997). Mechanism of helix induction by trifluoroethanol: A framework for extrapolating the helix-forming properties of peptides from trifluoroethanol/water mixtures back to water. Biochemistry, 36(27), 8413–8421. https://doi.org/10.1021/bi9707133

2.      The authors report using locally weighted regression fitting to try to understand the relationship between properties and their observed MIC.  Is it correct to presume that they attempted linear regression first but that it was not as revealing?

3.      The reviewer is curious about the author’s logic for their choice of developing the CM peptide series.  For CM1, for example, why did they choose to make the G4K alteration first?  Did they have some knowledge of why this would be better as opposed to creating CM1 as a S15K (furthers from the proline) or even putting it in the middle as a G8K?

Round 2

Reviewer 1 Report

Comments and Suggestions for Authors

I appreciate that the authors have abandoned the MIC averages, but I still have my doubts about the correlations shown in Figure 3.

Nevertheless, the overall results are of high quality and, in my opinion,  deserve to be published in Toxins.

Author Response

I appreciate that the authors have abandoned the MIC averages, but I still have my doubts about the correlations shown in Figure 3.

Nevertheless, the overall results are of high quality and, in my opinion,  deserve to be published in Toxins.

Reply: Thank you. The correlation presented in Figure 3 is a kind of interpretation of the data in Table 2, which is based on our designed peptides. It is not a universal rule.